Palaeogeographic implications of a new iocrinid crinoid (Disparida) from the Ordovician (Darriwillian) of Morocco

Zamora Samuel 1 s.zamora@igme.es samuel@unizar.es
Rahman Imran A. 2
Ausich William I. 3
1 Instituto Geológico y Minero de España , Zaragoza , Spain
2 School of Earth Sciences, University of Bristol , Bristol , United Kingdom
3 School of Earth Sciences, Ohio State University , Columbus, OH , United States
Kraatz Brian
Electronic publication date: 2015 Dec 7
Publication date: 2015
Volume: 3
Electronic Location ID: e1450
Received 2015 Sep 4; Accepted 2015 Nov 6
Copyright: © 2015 Zamora et al.
Copyright year: 2015
Copyright holder: Zamora et al.
License: This is an open access article distributed under the terms of the Creative Commons Attribution License, which permits unrestricted use, distribution, reproduction and adaptation in any medium and for any purpose provided that it is properly attributed. For attribution, the original author(s), title, publication source (PeerJ) and either DOI or URL of the article must be cited.
License URL: https://creativecommons.org/licenses/by/4.0/

Keywords: Crinoidea, Paleozoic, Micro-CT, Paleogeography, Morocco, Ordovician

Funding: Ramón y Cajal RYC-2012-10576 Spanish Ministry of Economy and Competitiveness CGL2012-39471 CGL2013-48877 1851 Royal Commission Research Fellowship National Science Foundation project Assembling the Echinoderm Tree of Life DEB 1036416 Samuel Zamora is funded by a Ramón y Cajal Grant (RYC-2012-10576) and projects CGL2012-39471 and CGL2013-48877 from the Spanish Ministry of Economy and Competitiveness. Imran Rahman is funded by an 1851 Royal Commission Research Fellowship. William I. Ausich is supported by the National Science Foundation project, Assembling the Echinoderm Tree of Life (DEB 1036416). The funders had no role in study design, data collection and analysis, decision to publish, or preparation of the manuscript.

==============================
Complete, articulated crinoids from the Ordovician peri-Gondwanan margin are rare. Here, we describe a new species, Iocrinus africanus sp. nov., from the Darriwilian-age Taddrist Formation of Morocco. The anatomy of this species was studied using a combination of traditional palaeontological methods and non-destructive X-ray micro-tomography (micro-CT). This revealed critical features of the column, distal arms, and aboral cup, which were hidden in the surrounding rock and would have been inaccessible without the application of micro-CT. Iocrinus africanus sp. nov. is characterized by the presence of seven to thirteen tertibrachials, three in-line bifurcations per ray, and an anal sac that is predominantly unplated or very lightly plated. Iocrinus is a common genus in North America (Laurentia) and has also been reported from the United Kingdom (Avalonia) and Oman (middle east Gondwana). Together with Merocrinus, it represents one of the few geographically widespread crinoids during the Ordovician and serves to demonstrate that faunal exchanges between Laurentia and Gondwana occurred at this time. This study highlights the advantages of using both conventional and cutting-edge techniques (such as micro-CT) to describe the morphology of new fossil specimens.

Introduction

Ordovician crinoids from west peri-Gondwana (North Africa and southwestern and central Europe) are relatively rare, with only a few species reported from Spain, France, Italy, Morocco, Portugal, and the Czech Republic (Ubaghs, 1969; Ubaghs, 1983; Prokop & Petr, 1999; Ausich, Gil Cid & Domínguez Alonso, 2002; Ausich, Sá & Gutiérrez-Marco, 2007; Correia & Loureiro, 2009; Zamora, Colmenar & Ausich, 2014; Sumrall et al., 2015). Crinoids from Morocco include an incomplete specimen assigned to Ramseyocrinus sp. by Donovan & Savill (1988) from the Upper Fezouata Formation, which is Floian (Early Ordovician) in age (sensu Ausich, Sá & Gutiérrez-Marco, 2007), and several well-preserved complete specimens of Rosfacrinus robustus (Le Menn & Spjeldnaes, 1996), from the Upper Tiouririne Formation (Lefebvre et al., 2007), which is Katian (Late Ordovician).

Most of the crinoid genera from the Ordovician of peri-Gondwana are endemic, and this hampers our ability to understand the migration patterns of crinoids during this important time interval, in which several echinoderm classes reached major peaks in diversity (Guensburg & Sprinkle, 2000; Sprinkle & Guensburg, 2004; Nardin & Lefebvre, 2010; Lefebvre et al., 2013). Until now, the only exception was Merocrinus, which has been reported from England (Avalonia), Spain (peri-Gondwana), and North America (Laurentia) (Ausich, Gil Cid & Domínguez Alonso, 2002). Herein, we report a new species of Iocrinus from the Ordovician of Morocco, thereby extending the range of this genus with certainty to encompass west peri-Gondwana (in addition to Avalonia and Laurentia; Donovan et al., 2011) and confirming its cosmopolitan distribution. Iocrinus africanus sp. nov. is described based on a single well-preserved specimen, which was collected from south Alnif (eastern Anti-Atlas, Morocco) and is preserved in a concretion found in the Taddrist Formation, which is Darriwilian in age (Rábano, Gutiérrez-Marco & García-Bellido, 2014). The new crinoid was studied using both traditional techniques (casting the mould in latex) and X-ray micro-tomography (micro-CT). This allows us to describe the morphology of Iocrinus africanus sp. nov. in great detail and serves as a basis for comparison with other species of Iocrinus.

Geological setting and stratigraphy

Ordovician outcrops are very well developed and exposed in the Anti-Atlas Mountains of Morocco (Destombes, Hollard & Willefert, 1985). Many units yield well-preserved specimens of echinoderms, a number of which are currently under study (e.g., Hunter et al., 2010; Van Roy et al., 2010; Van Roy, Briggs & Gaines, 2015; Sumrall & Zamora, 2011; Martin et al., in press), and these faunas occur throughout sections from the Lower to Upper Ordovician. Numerous clades of echinoderms have been documented, including stylophorans, solutes, blastozoans, crinoids, asteroids, edrioasteroids, and cyclocystoids.

The Ordovician succession in the Anti-Atlas region is divided into the following lithostratigraphic units: the Outer Feijas Shale Group, the First Bani Sandstone Group, the Ktaoua Clay and Sandstone Group, and the Second Bani Sandstone Group (Choubert, 1943; Choubert & Termier, 1947; Destombes, Hollard & Willefert, 1985). The Outer Feijas Shale Group includes the Lower and Upper Fezouta formations (Tremadocian–Floian) and the Tachilla Formation (Darriwilian) (Fig. 1). These units are characterized by siltstones that are rich in graptolites, with some thin sandstone interbeds, and contain exceptionally preserved Burgess Shale-type faunas in places (Van Roy et al., 2010; Van Roy, Briggs & Gaines, 2015; Martin et al., in press). The overlying First Bani Group spans the Darriwilian to Sandbian and is subdivided into five formations (Taddrist, Bou-Zeroual, Guezzart, Ouine-Inirne, and Izegguirene formations) that are chiefly comprised of sandstones with interbedded shales. This group is the thickest, most constant, and most extensive sandstone group in the Anti-Atlas Mountains (Destombes, Hollard & Willefert, 1985). The fossil taxa recovered from the First Bani Group were reviewed by Gutiérrez-Marco et al. (2003), and there are no reports of crinoids from this time interval.

Figure 1 Chronostratigraphical chart for the Ordovician, indicating the levels that provided the studied specimen.

Correlations between stratigraphical units in the Anti-Atlas (after Destombes, Hollard & Willefert, 1985; Gutiérrez-Marco et al., 2003; Villas et al., 2006), the British regional time scale (Fortey et al., 1995), North American graptolite zonal sequences (Webby et al., 2004), Mediterranean regional stages (Gutiérrez-Marco et al., 2003), and global stages are shown. Modified from Sumrall & Zamora (2011). Abbreviations: Kral, Kralodvorian; pars., partial; Tr., Tremadocian.

The First Bani Group is overlain by the Ktaoua Clay and Sandstone Group (Sandbian–Katian), which is comprised of siltstones interbedded with two or three sandstone units, depending on the exact position within the Anti-Atlas Mountains. It is divided into three units: the Sandbian to Katian Lower Ktaoua Formation, the Katian Upper Tiouririne Formation, and the Katian Upper Ktaoua Formation. The Ordovician ends with the Second Bani group, which is Hirnantian in age.

The new locality yielding Iocrinus africanus sp. nov. lies in the Taddrist Formation, low in the First Bani Group, close to the village of Battou (south Alnif, eastern Anti-Atlas) (Figs. 1 and 2). This locality was recently described by Rábano, Gutiérrez-Marco & García-Bellido (2014), who provided detailed information about the faunal content and age based on the presence of key graptolites and trilobites. In this area, the Taddrist Formation has been excavated predominantly by local collectors and has yielded a rich faunal assemblage preserved in carbonate concretions (Fig. 3). Rábano, Gutiérrez-Marco & García-Bellido (2014) suggested that the levels containing fossiliferous concretions belong to the Didymograptus murchisoni Biozone (Gutiérrez-Marco et al., 2003), which is assigned to the upper Oretanian, a regional stage roughly equivalent to the upper Darriwilian 2/basal Darriwilian 3 stage slices of the global chronostratigraphic scale (Gutiérrez-Marco, Sá & Rábano, 2008; Bergström et al., 2009). According to Rábano, Gutiérrez-Marco & García-Bellido (2014), the fossiliferous concretions have yielded the trilobites Caudillaenus nicolasi (Rábano, Gutiérrez-Marco & García-Bellido, 2014), Morgatia? rochi (Destombes, 1972), Placoparia (Coplacoparia) sp. nov., Colpocoryphe sp., Parabarrandia aff. crassa (Barrande, 1872), and an undetermined cheirurid (Eccoptochile? sp.). Other non-trilobite fossils include molluscs (e.g., a cyrtonellid tergomyan, bivalves such as Praenucula sp., and orthoconicnautiloids), hyoliths (Elegantilites sp.), echinoderms (Diploporita and Asterozoa indet.), conularids (Exoconularia sp.), and rare graptolites (Didymograptus sp.). In addition to the crinoid described herein, new cyclocystoids, the first ever reported from Africa, were recently presented from this locality and await formal description (Sprinkle, Reich & Lefebvre, 2015).

Figure 2 Geographical and geological setting of the eastern Anti-Atlas Mountains, Morocco, showing the type locality of the new species (indicated by a star) close to the village of Battou.

After Rábano, Gutiérrez-Marco & García-Bellido (2014). (A) Map of Africa. (B) Detailed map of west Africa showing the position of the Anti-Atlas Mountains. (C) Simplified geological map of Morocco with the position of the studied locality; a: Precambrian and Palaeozoic rocks, b: Ordovician rocks, c: post-Palaeozoic cover. (D) Geographic map indicating the position of the studied locality.

Figure 3 Field photographs showing the Taddrist Formation and the levels yielding fossiliferous concretions.

(A) General view of the Taddrist Formation in the studied area. (B) Detail of the trench providing the fossiliferous concretions.

Material and Methods

The studied specimen is preserved in a yellowish carbonate concretion that is approximately 70 mm in length and 45 mm in width. The crinoid is preserved as a natural mould and includes the complete theca, articulated arms, and part of the column. The specimen is housed in the Museo Geominero (Madrid, Spain) under the repository number MGM 6754.

A latex cast of the specimen was prepared to study the morphology of the animal (Fig. 4). In addition, the specimen was imaged using micro-CT and digitally reconstructed to characterize the fossil in three dimensions (Fig. 5). The specimen was scanned on a Nikon XT H 225 cabinet scanner at the Natural History Museum, London with a 0.5 mm thick copper filter, 215 kV voltage, 177 µA current, and 3,142 projections (each with an exposure time of 708 ms). Tomographic reconstruction was performed in Nikon CT Pro software using filtered back projection, giving a tomographic dataset with a voxel size of 37 µm. This dataset was then visualized with the free SPIERS software suite (Sutton et al., 2012); an inverted linear threshold was applied to the dataset, and the pixels that could be unambiguously identified as representing the crinoid were manually assigned to a separate region-of-interest. Isosurfaces were rendered to give an interactive three-dimensional model of the fossil, which was subjected to weak smoothing and island removal to reduce noise. Micro-CT slices, segmented images, and the interactive 3-D model (in VAXML format) are provided at the following DOI http://dx.doi.org/10.5523/bris.uv7qt4c6kpat1befj0937ooml.

Figure 4 Iocrinus africanus sp. nov. (MGM 6754) from the Darriwilian (Middle Ordovician) of Morocco.

(A), (B) General morphology including the complete crown showing the E-ray (A) and BC-interray (B), the proximal column, and part of the arms. (C) Detail of the cup showing the E-ray. (D) Detail of the cup showing the A-ray. (E) Detail of the cup showing the D-ray. All images are photographs of latex casts of the specimen whitened with ammonium chloride sublimate.

Figure 5 Iocrinus africanus sp. nov. (MGM 6754) from the Darriwilian (Middle Ordovician) of Morocco.

Digital reconstructions of the specimen. (A) General morphology showing the AE-interray. (B) Detail of the theca showing the C-ray. (C) Detail of the cup showing the BC-interray. (D) Detail of the cup showing the D-ray. (E) Detail of the column showing pentastellate shape and holomeric construction. (F) Detail of the proximal arms showing the E-ray. (G) Column in an open coil. Abbreviations: A–E, ambulacra.

Terminology

The terminology used below follows Moore (1962), Webster (1974), Ubaghs (1978), and Ausich et al. (1999); the classification follows Ausich (1998). Note, the terminology used for the aboral plates differs from that of Ausich, Gil Cid & Domínguez Alonso (2002). In addition, superradial and inferradial are used to designate radially positioned plates where two plates are in the C ray portion of the radial circlet. This usage recognizes the homologies of these plates (essential for phylogenetic analysis) rather than using unique names that obscure homology, such as anibrachial and brachianal as outlined in Ubaghs (1978). The present usage is consistent with many recent studies (e.g., Ausich, 1998; Ausich & Copper, 2010; Ausich et al., 2015; Ausich, Rozhnov & Kammer, 2015), although the Ubaghs (1978) terminology for these plate is also used (e.g., Guensburg, 2010).

Nomenclatural acts

The electronic version of this article in Portable Document Format (PDF) will represent a published work according to the International Commission on Zoological Nomenclature (ICZN), and hence the new names contained in the electronic version are effectively published under that Code from the electronic edition alone. This published work and the nomenclatural acts it contains have been registered in ZooBank, the online registration system for the ICZN. The ZooBank LSIDs (Life Science Identifiers) can be resolved and the associated information viewed through any standard web browser by appending the LSID to the prefix “http://zoobank.org/”. The LSID for this publication is: http://zoobank.org/References/3F137EAC-ECB5-4FF8-9BA5-8B1A0E1BC986. The online version of this work is archived and available from the following digital repositories: PeerJ, PubMed Central and CLOCKSS.

Results

Systematic Paleontology

Class CRINOIDEA Miller, 1821	
Subclass DISPARIDA Moore & Laudon, 1943	
Order MYELODACTYLIDA Ausich, 1998	
Family IOCRINIDAE Moore & Laudon, 1943	
Genus Iocrinus Hall, 1866	

Type species

Heterocrinus (Iocrinus) polyxo Hall, 1866 = Heterocrinus subcrassus Meek & Worthen, 1865.

Diagnosis

Iocrinid with basal plates visible in lateral view; anal sac with large plicate plates if calcified; variable number of primibrachials; arms branch as many as eight times; fixed interradial plates absent; column holomeric, pentalobate throughout; columnal facets in mesistele petaloid.

Iocrinus africanus sp. nov. urn:lsid:zoobank.org:act:D091338E-643F-4D5A-8A08-7D7D190DBC2E.

Holotype

MGM 6754, a nearly complete, articulated specimen not retaining the attachment structure and dististele preserved as a mould in a carbonate concretion (Figs. 4 and 5; Data S1, S2, http://dx.doi.org/10.5523/bris.uv7qt4c6kpat1befj0937ooml).

Type locality and age

Close to the village of Battou, south Alnif, eastern Anti-Atlas, Morocco (Fig. 2); Taddrist Formation, Darriwilian (Middle Ordovician).

Etymology

Named in reference to the African continent.

Diagnosis

Basal plate height approximately 37 percent of radial plate height; radial plates 1.25 times higher than wide; single, broad transverse ridge between adjacent radial plates; primibrachials 1.5 times wider than high; three to five primibrachials; four to five secundibrachials; seven to thirteen tertibrachials; three in-line bifurcations per ray; anal sac unplated or very lightly plated (except for the robust column of plates from the C-ray superradial); proximal columnals pentastellate.

Description

Crown small in size. Aboral cup medium bowl-shaped; smooth plate surfaces; radial and basal plates sharply convex.

Basal circlet 27 percent of aboral cup height; five basal plates, approximately two times wider than high, much smaller than radial plates. Radial circlet 73 percent of aboral cup height; radial plates five, maximum height approximately 1.25 times higher than maximum width; maximum width of radial plate at mid-height, radials narrow sharply proximally, maximum width more than 10 times proximal width; maximum width 1.6 times distal width. Radial facets peneplenary, approximately as deep as wide. A, B, D, E radial plates simple, C radial compound; C inferradial approximately same size as simple radials; C superradial much smaller than C inferradial, wider than high, distal heterotomous division with anal plates to left and C-ray arm to right.

All anal plates above aboral cup; column of 16 stout anal sac plates preserved from the left facet on the C-ray superradial, plates very convex, successive plates with bend yielding a sinuous appearance for this column of plates; each plate higher than wide, otherwise very similar to shape of brachials. Other anal sac plates disarticulated and collapsed within the crown, presumably sac plates were lightly calcified or uncalcified, except for the column of plates from the C superradial.

Arms robust, primaxil varies from third to fifth primibrachial (45553; ABCDE), secundaxil fourth or fifth secundibrachial; where known, tertaxil positioned on the seventh or thirteenth tertibrachial; as many as 16 unbranched quartibrachials on an incomplete branch of the A-ray arm. Brachials strongly convex aborally with flattened lateral, abambulacral extensions, rectangular uniserial, deep ambulcacral groove, more proximal brachials approximately 1.7 times wider than high. Brachial facet with two, merging aboral ligament fossae. Primaxial approximately the same size as non-axillary primibrachials; remaining brachials diminish in size distally.

Column strongly pentastellate, holomeric, heteromorphic, proximal column N3231323; nodals higher than priminternodals, obvious heteromorphic pattern lacking in mesistele, large portion of columnal facets presumably a petaloid articulation (but details not preserved). Preserved column higher than crown height and preserved in an open coil.

Remarks

Characters differentiating genera within the Iocrinidae are listed in Ausich, Rozhnov & Kammer (2015). The combination of visible basal plates, three to five primibrachials, no fixed interradial plates, pentalobate/pentastellate columnal shape, holomeric column construction, and a petaloid facet clearly align the new crinoid described herein with the genus Iocrinus. Another feature that identifies the specimen as belonging to Iocrinus is the preservation of the column in an open coil. This is similar to Iocrinus subcrassus, which is thought to have had a holdfast that could coil around erect objects (Kelly, 1978; Brett, Deline & McLaughlin, 2008; Meyer & Davis, 2009).

Species-level characters within Iocrinus include: the height of the basal plates, the height of the radial plates, radial plate height versus width, presence and character of the transverse ridge between adjacent radial plates, primibrachial shape, number of primibrachials, number of secundibrachials, number of tertibrachials, maximum number of in-line bifurcations in a ray, anal sac plating, and the shape of the proximal columnal (Table 1). Iocrinus africanus sp. nov. is distinguished from other Iocrinus species based on the shape of the radial plates, the number of tertibrachials, the number of bifurcations in-line per ray, and the lack of or very light plating of most of the anal sac.

Table 1 Morphological comparison of Iocrinus species.

Iocrinus species	Basal plate height	Radial plate height vs. width	Transverse ribbing on between adjacent radial plates	Primibrachial shape	Number of primibrachials	Number of secundibrachials	Number of tertibrachials	Number of arm bifurcations in line	Anal Sac plating robust	Proximal column shape	
Iocrinus crassus	Approximately 50% of radial plate height	Height approximately equals width	Yes, single, broad	2.0 times wider than high	4 to 5	4 to 6	4 to 8	As many as 7	Unknown	Pentastellate	
Iocrinus llandegleyi	Approximately 67% of radial plate height	Slightly wider than high	No	2.0 times wider than high	5 to 8	4 to 5	4 to 5	At least 3	Yes	Pentastellate	
Iocrinus pauli	Approximately 60% of radial plate height	Height approximately equals width	Yes, double, narrow	Less than 2.0 times wider than high	5	5 to 6	5 to 8	4	Yes	Pentalobate	
Iocrinus similis	Unknown	Height approximately equals width	Unknown	1.5 times wider than high	3 to 4	Unknown	Unknown	Unknown	Unknown	Unknown	
Iocrinus subcrassus a	Approximately 50% of radial plate height	Height less than width	Yes, single, narrow	2.0 times wider than high	3 to 8	4 to 5	5 to 13	Typically 4 but can be 3 to 8	Yes	Pentalobate	
Iocrinus subcrassus torontoensis	Approximately 50% of radial plate height	Height approximately equals width	Yes, single, narrow	2.0 times wider than high	5	6 to 7	6 to 11	4	Yes	Pentastellate?	
Iocrinus trentonensis	Approximately 50% of radial plate height	Height approximately equals width	Yes, single, broad	1.5 times wider than high	4 to 6	6 to 9	>12	4	Yes	Pentalobate?	
Iocrinus whitteryi	Approximately 67% of radial plate height	Slightly wider than high	No	More than 2.0 times wider than high	7	Unknown	Unknown	Unknown	Yes	Unknown	
Iocrinus africanus n.sp.	Approximately 37% of radial plate height	1.25 times higher than wide	Yes, single, broad	1.5 times wider than high	3 to 5	4 to 5	7 to 13	3	No	Pentastellate	
Notes.

Diagnostic table for species of Iocrinus

a indicates type species

Donovan et al. (2011) reported the only other putative Iocrinus known from Gondwana, I. sp. cf. I. subcrassus from the Middle Ordovician of Oman. Assuming that this taxon does belong to Iocrinus, which cannot be confirmed without further information about the CD-interray and C-ray morphologies, the new Moroccan species differs from the Donovan et al. (2011) specimen as follows. Iocrinus africanus sp. nov. has a basal plate height approximately 37 percent of radial plate height; a broad transverse ridge; primibrachials 1.5 times wider than high; four to five secundibrachials; and three in-line bifurcations per ray. In contrast, I. sp. cf. I. subcrassus has a basal plate height approximately 50 percent of radial plate height; a narrow transverse ridge; primibrachials slightly higher than wide; seven secundibrachials; and as many as seven in-line bifurcations per ray.

Taxonomic assignments within the Iocrinidae have received some attention in the last three decades (Warn, 1982; Guensburg, 1984; Donovan, 1985; Donovan, 1989; Ausich, Rozhnov & Kammer, 2015); with the new species described herein, a total of eight species and one subspecies are currently recognized for Iocrinus (Webster & Webster, 2014). These include the Laurentian species: I. crassus (Meek & Worthen, 1865); I. similis (Billings, 1865); I. subcrassus (Meek & Worthen, 1865); I. subcrassus torontoensis (Fritz, 1925); and I. trentonensis (Walcott, 1884); and the Avalonian species: I. llandegleyi (Botting, 2003); I. pauli (Donovan & Gale, 1989); and I. whitteryi (Ramsbottom, 1961) (Table 2). Additional Iocrinus identifications left in open nomenclature are known from Avalonia, Laurentia, and Gondwana (for the previous potential Gondwanan occurrence, see Donovan et al., 2011). Iocrinus africanus sp. nov. is Darriwilian in age, and thus it is among the oldest members of the genus (Table 2). In terms of morphology, it is equally dissimilar to species from both Laurentia and Avalonia. The occurrence of I. africanus sp. nov. in Morocco confirms the presence of Iocrinus in Gondwana and demonstrates that Iocrinus, together with Merocrinus, is the most geographically widespread Ordovician crinoid genus.

Table 2 Stratigraphic and geographical distribution of species of Iocrinus and Merocrinus.

Genus	Species	Formation	Age	Location	Country	Paleo-continent	
IOCRINUS	
	Iocrinus sp. cf. I. subcrassus	Amdeh Formation	late Dapingian or early Darriwilian	Muscat	Oman	Gondwana	
	Iocrinus llandegleyi	Builth Volcanic Group	Darriwillian	Wales	UK	Avalonia	
	Iocrinus pauli	Camnant Mudstone	Darriwillian	Wales	UK	Avalonia	
	Iocrinus pauli	Didmograptus bifidus Beds	Darriwillian	England	UK	Avalonia	
	Iocrinus sp. cf. pauli	Llandeilo Flags	Darriwillian	Wales	UK	Avalonia	
	Iocrinus cf. whitteryi	volcanic sandstones	Darriwillian	England	UK	Avalonia	
	Iocrinus whitteryi	Chirbury Formation	Sandbian	England	UK	Avalonia	
	Iocrinus cf. subcrassus	Whittery Beds	Sandbian	England	UK	Avalonia	
	Iocrinus subcrassus	Arnheim Formation	Katian	Southwestern Ohio Region	USA	Laurentia	
	Iocrinus subcrassus	Lorraine Shale	Katian	New York	USA	Laurentia	
	Iocrinus cf. subcrassus	Lorraine Shale	Katian	New York	USA	Laurentia	
	Iocrinus subcrassus	Cobourg Limestone	Katian	Ontario	Canada	Laurentia	
	Iocrinus subcrassus	Georgian Bay Formation	Katian	Ontario	Canada	Laurentia	
	Iocrinus subcrassus torontoensis	Dundas Formation	Katian	Ontario	Canada	Laurentia	
	Iocrinus similis	Cobourg Limestone	Katian	Ontario	Canada	Laurentia	
	Iocrinus subcrassus	Correyville Formation	Katian	Southwestern Ohio Region	USA	Laurentia	
	Iocrinus sp.	Fort Atkinson Formation	Katian	Iowa and Illinois	USA	Laurentia	
	Iocrinus subcrassus	Fairview Formation	Katian	Southwestern Ohio Region	USA	Laurentia	
	Iocrinus sp.	Kope Formation	Katian	Southwestern Ohio Region	USA	Laurentia	
	Iocrinus subcrassus	Liberty Formation	Katian	Southwestern Ohio Region	USA	Laurentia	
	Iocrinus crassus	Maquoketa Shale	Katian	Illinois	USA	Laurentia	
	Iocrinus trentonensis	Rust Formation	Katian	New York	USA	Laurentia	
	Iocrinus trentonensis	Trenton Limestone	Katian	New York	USA	Laurentia	
	Iocrinus subcrassus	Waynesville Formation	Katian	Southwestern Ohio Region	USA	Laurentia	
MEROCRINUS	
	Merocrinus millanae	Guindo Shales	Darriwilian	Embalse de Fresneda	Spain	Gondwana	
	Merocrinus salopioe	Meadowtown Beds	Darriwilian		England	Avalonia	
	Merocrinus britonensis	Mifflin Formation	Sandbian	Illinois	US	Laurentia	
	Merocrinus britonensis	Platteville Group	Sandbian	Illinois, Iowa, Wisconsin, Minn	US	Laurentia	
	Merocrinus impressus	Bromide Formation (Pooleville Mbr.)	Sandbian	Oklahoma	US	Laurentia	
	Merocrinus impressus	?	?	?	Sweden	Baltica	
	Merocrinus curtus	Kope Formation	Katian	Southwestern Ohio Region	US	Laurentia	
	Merocrinus curtus	Rust Formation	Katian	New York	US	Laurentia	
	Merocrinus retractilis	Rust Formation	Katian	New York	US	Laurentia	
	Merocrinus sp.	Wisf Formation (Sinsinewa Mbr.)	Katian	Illinois and Iowa	US	Laurentia	
	Merocrinus corroboratus	Trenton Limestone	Katian	New York	US	Laurentia	
	Merocrinus typus	Trenton Limestone	Katian	New York	US	Laurentia	

The use of micro-CT was essential for describing the morphology of Iocrinus africanus sp. nov. in full. The posterior interray is buried below the surface of the concretion and is hence not visible in the latex casts (Fig. 4); however, the posterior interray and the C-ray can be clearly seen in the micro-CT scans (Fig. 5; Data S1, S2, http://dx.doi.org/10.5523/bris.uv7qt4c6kpat1befj0937ooml). Without an understanding of these characters, it would not have been possible to confidently assign the specimen to the genus Iocrinus.

Paleobiogeographical Implications

The Middle to Late Ordovician was characterized by high degrees of endemism in crinoids (Paul, 1976; Lefebvre et al., 2013), and Iocrinus and Merocrinus are the only geographically widespread genera from this period (Fig. 6). Both genera first appeared in Gondwana and/or Avalonia during the Darriwillian. Merocrinus first occurred in Laurentia during the Sandbian, and Iocrinus first occurred in Laurentia during the Katian (however, Sprinkle, Guensburg & Gahn, 2008 noted the occurrence of older, undescribed iocrinids and a merocrinid-like cladid? from faunas in North America). Based on presently described taxa, the known geographical distribution of these genera indicates that their migration to Laurentia was asynchronous. Iocrinus is a disparid crinoid, and disparids are usually recognized as having a more widespread geographic distribution and temporal range than other clades (Kammer, Baumiller & Ausich, 1998). Merocrinus is generally considered to be a cladid (but see Sprinkle & Guensburg, 2013), which in general are not as cosmopolitan as disparids, at least later during the Paleozoic. Unfortunately, there is not currently enough known about the life history of Paleozoic crinoids to propose any explanation for the cosmopolitan nature of Iocrinus and Merocrinus during the Ordovician.

Figure 6 Distribution of the major paleocontinents during the Middle Ordovician, showing the known geographical distribution of Iocrinus and Merocrinus.

Locality markers indicate the presence of a taxon on a palaeocontinent; multiple localities are not noted on a single palaeocontinent. Modified from Cocks & Torsvik (2006).

We appreciate the comments of four reviewers, Christian Klug (University of Zurich), Stephen Donovan (Naturalis Biodiversity Center), James Sprinkle (University of Texas at Austin) and Forest Gahn (Brigham Young University), which greatly improved the final manuscript. We thank Isabel Pérez (University of Zaragoza) for providing photographs of the studied specimen and Dan Sykes (Natural History Museum, London) for assistance with micro-CT. The specimen was collected by Samuel Zamora on a field trip accompanied by Andrew Smith (Natural History Museum, London). Juan Carlos Gutiérrez-Marco (Spanish Research Council) and Richard Fortey (Natural History Museum, London) provided helpful comments on the associated fauna, and Bertrand Lefebvre (University of Lyon) is thanked for important discussions on echinoderms from Morocco.

Additional Information and Declarations

Competing Interests

Author Contributions

Data Availability

New Species Registration

The authors declare there are no competing interests.

Samuel Zamora, Imran A. Rahman and William I. Ausich conceived and designed the experiments, performed the experiments, analyzed the data, contributed reagents/materials/analysis tools, wrote the paper, prepared figures and/or tables, reviewed drafts of the paper.

The following information was supplied regarding data availability:

Palaeogeographic implications of a new iocrinid crinoid (Disparida) from the Ordovician (Darriwillian) of Morocco DOI http://dx.doi.org/10.5523/bris.uv7qt4c6kpat1befj0937ooml.

The following information was supplied regarding the registration of a newly described species:

Iocrinus africanus Zamora, Rahman & Ausich: urn:lsid:zoobank.org:act:D091338E-643F-4D5A-8A08-7D7D190DBC2E

Publication LSID: http://zoobank.org/References/3F137EAC-ECB5-4FF8-9BA5-8B1A0E1BC986.

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
