# Peer review of "Palaeogeographic implications of a new iocrinid crinoid (Disparida) from the Ordovician (Darriwillian) of Morocco"

_PeerJ, doi:10.7717/peerj.1450_

## Round 0.1 · original submission · Minor Revisions

Dear Samuel -

I am recommending 'minor revisions' for your thorough and well written study. You will see that all four reviewers were in agreement on the strength of this manuscript, and the overall quality. Please be sure to respond to all reviewer comments. My only addition is that it doesn't seem you defined MGM, or reference specifically where these specimens will be permanently accessioned. Please clarify this.

I'd also recommend that you choose to make this an 'open' review to support the transparency of our review process; over 80% of articles published in PeerJ choose this option.

I appreciate the degree to which you are also making these data accessible to future researchers. Please finalize a permanent depository for supplemental files.

Once I have your response, I anticipate a final decision in short order.

This is a fine piece of science, thank you for choosing PeerJ.

·

Basic reporting

The English is excellent, the illustrations adequate and of high quality and the entire ms is concise. There are only a few very minor issues I point out in the annotated pdf.
The authors should consider mentioning their palaeogeographic findings and perhaps also the use of CT in the title of the paper. As it is, it sounds very systematic although there is a chpater on palaeogeography.

Experimental design

There are no true experiments. However, the use of CT-data is highly professional and appears adequate to me.

Validity of the findings

No complaints.

Additional comments

Congratulations to this concise and beautifully illustrated manuscript!

·

Basic reporting

This is a well-written report of a new species in one of my favourite taxa (I've been working on iocrinids for 35 years!). Well worth reporting, particularly supported by the palaeobiogeographic discussion. Any comments I have made on the pdf are minor and will easily be dealt with by the authors. It satisfies all of your criteria (right).

Experimental design

Satisfactory.

Validity of the findings

Satisfactory.

·

Basic reporting

Manuscript is well written, introductory sections are very good with wider implications, has well delineated template sections for a paleontological manuscript, has very good figures with good descriptions, is a nicely done self-contained description of a new fossil crinoid species from a new area in Gondwana, and has the raw data and viewing model are available in 2 supplementary data files.

Experimental design

Manuscript describes a new fossil crinoid species from early in the fossil record found in a poorly known area (NW Gondwana), study fills several gaps in knowledge (diversity, paleogeography new way of studying mould specimens), has been conducted rigorously, methods are extensive, perhaps overly so, and MS conforms with ethical standards in paleontology.

Validity of the findings

The data in this manuscript appears robust, and are available in an open repository, conclusions are well stated and supported by the results, and there is very little speculation.

Additional comments

This is a very nice manuscript describing a new species of a fairly well-known fossil crinoid genus. The problems listed below are mostly minor and should be easy to fix:
1) Page 3, l. 57: should state what "traditional techniques" mean when 1st mentioned – "using traditional techniques (casting the mould in latex) and X-ray". . .
2) Page 6, l. 113: "yellowish carbonate concretion"– was this carbonate siderite, how thick was the siderite crust, & was the concretion already cracked open and weathering when found?
3) Page 7, line 132 & Page 9, line 189-190 – why is there no mention that an alternate terminology for C-ray anal plating in disparids exists (C radial & anibrachial, vs. "C radial compound; C inferradial . . .; C superradial. . ."), and is actually favored for iocrinids by Ubaghs (1978, cited), and by other recent authors describing very early Iocrinid-like disparids, such as Alphacrinus (Guensburg, 2010, not cited)?
4) Page 8, line 159 – Why is there no telegraphic Diagnosis for the Genus Iocrinus where first used in the Systematic paleontology section, instead of a longer, rambling, 5-line discussion sending readers to a hard-to-get Russian reference in the Remarks after the species Description (Page 10, lines 210-214)? This move would put the diagnostic information where expected, and might save several lines of text.
5) Page 13, line 274 vs. Page14, line 300 – Zamora is mentioned as collecting the one studied specimen on line 274, but not in the Author Contributions section, which is a rather important omission! Should change line 300 to read "Samuel Zamora collected the holotype specimen, conceived and designed . . ."
6) Page13, lines 279-284 & Page 14, lines 285-294 – Combine these 2 sections as "Funding and Grant Disclosures" with lines 279-286; delete lines 289-294 (saves 6 lines).
7. Page 16, line 344 (Destombes reference) – "de l'Ordovicien" (delete extra space).
8. Page 17, between lines 368 & 369 – Shouldn't authors cite "Guensberg TE. 2010. Alphacrinus new genus and origin of the disparid clade. Journal of Paleontology 84:1209-1216. which could be cited on p. 7, line 132 or elsewhere as the oldest known iocrinid-like disparid crinoid?
9) Page 19, line 409 – Shouldn't the 4 genera cited in this italicized book title be Roman to set them off from the rest of the title?
10) Page 20, between lines 432 & 433 – Might want to add "Sprinkle, J, Guensburg, TE, Gahn, FJ. 2008. Overview of Early Ordovician crinoid diversity from the western and southwestern United States. In: Ausich WI, Webster, GD, eds. Echinoderm Paleobiology. Bloomington: Indiana University Press. 312-328." based on comment 18 below.
11) Page 21, line 441 –"Siveter DJ, Siveter DJ,"; one of these is David and the other is Derek; is there some way of indicating which is which in this journal's format?
12) Page 21, lines 452-453 "Echinodermata 2(1). . . University of Kansas." (no Press)
13) Page 21, line 457 – "delete long dash in "–Early Ordovician."?
14) Page 22, line 486 – "showing the type locality of this new species (indicated"; also, what are a, b, & c in this figure? (delete?)
15) Page 23, line 489 – would trench indicate there was only one "level"? What came from the other "levels"?
16) Page 23, lines 493-495 & 501-502 – I'm bothered by use of 'theca" vs. crown or cup here; is this new terminology? Rays A-E are present on all parts of this iocrinid (the cup, arms, and even the stem), not just the "theca".
17) Page 23, line 504 – "Black abbreviations A-E on cup: rays with ambulacra."?
18) Page 29, Figure 6 – You have the red dot for this new Moroccan crinoid on the wrong continent in Gondwana (NW South America vs. NW Africa)!!! Move red dot NE to just west of South Pole. Also in Figure 6, you might want to put a red-green dot in southern Oklahoma (near the "L"in Laurentia) for a merocrinid and iocrinids in the Arbuckle Group (Merocrinus S.S. also occurs in the Bromide), and one or more red dots on the extreme left edge of Laurentia for Iocrinids in S. Idaho, W. Utah, & central Nevada mentioned in Sprinkle, Guensburg, and Gahn, 2008. All of these central and western Laurentian occurrences are Floian (late Early Ordovician), but we're not yet sure these are strictly Iocrinus and Merocrinus. If true, this would mess up your paleobiogeographic conclusions on pages 12-13.

·

Basic reporting

The manuscript is very well written. In fact, it is one of the most clearly written manuscripts that I've reviewed recently. The figures are exceptional, but I do have one suggestion on this front. Unless I am mistaken, this new species is based on a single specimen. Figure 1 provides a "range" for this species. Single specimens can't have a range; thus, I suggest using an arrow or a symbol other than a bar (which infers a range) and titling the symbol something other than "stratigraphic range" (stratigraphic position or occurrence, perhaps).

On line 35 of the ms, should "Southwestern" and "Central" be lower case instead of capitalized?

Experimental design

The purpose of the research is very clearly articulated. On the surface it appears to be a paper that simply describes a new species of Iocrinus, but it is much richer than that. This new species is placed into a paleogeographical context. Moreover, the description of the specimen is greatly enhanced by micro-CT. This is a wonderful example to follow for specimens preserved as molds.

Validity of the findings

The new species of Iocrinus is very well documented. The diagnostic table for species of Iocrinus (Table 1) is very useful, but it also demonstrates how similar (and variable) many of these species appear to be. Although it's beyond the scope of the manuscript, it might be worth commenting on the validity of these species. At some point it would be worthwhile to generate a cladogram for the species of Iocrinus, but not necessarily in this manuscript. The addition of information on Merocrinus to this paper was a little surprising, but it's reasonable given that it is another example of a geographically widespread Ordovician crinoid. I don't think there is any mention of this in the manuscript, but I can't help but wonder if their widespread distribution isn't at least partially related to their preference for relatively deep water environments.

Additional comments

Congratulations on a well-written manuscript. It was very enjoyable reading one that was so polished. Moreover, the figures are very well done. Thanks for listing me as a reviewer. I wish I could offer more suggestions for improvement, but it's in great shape as it is.

---

## Round 0.2 · accepted · Accept

I caught one small error that I point out here so that you and the editors can correct it in the proofing stage.

Line 144: 'plates'